# Fast Block Attention Computation via Dynamic Algorithm

## Abstract

Recent progress in video modeling has been largely driven by Transformer architectures, which simulate dependency relationships across spatial patches and temporal frames. However, compared to text or image modeling, video modeling involves orders of magnitude more tokens, resulting in an input sequence several orders of magnitude longer than typical NLP or image tasks, and makes the attention mechanism the primary computational bottleneck. The naive method flattens $f$ frames of $n$ tokens each into length $N = nf$, incurring total $O(n^2 f^2)$ attention cost. Prior work (e.g., radial/axial variants) attains subquadratic time only when either the spatial or temporal dimension is small. We present a dynamic algorithm that computes block attention in $O(\mathcal{T}_{\mathrm{mat}}(n, n, n^a)\frac{f}{n^a})$ amortized running time, where $a \in [0, 1)$.

## 1 Introduction

Large Language models (LLMs) such as Transformer (Vaswani et al., 2017), BERT (Devlin et al., 2019a), PaLM (Chowdhery et al., 2023), and GPT-4o (Hurst et al., 2024) have demonstrated remarkable capabilities in natural language understanding and generation, which helps to achieve a wide range of tasks such as language translation, sentiment analysis, and question answering. Similarly, video Transformers such as TimeSformer (Bertasius et al., 2021), ViViT (Arnab et al., 2021), and Video Swin Transformer (Liu et al., 2022) have illustrated remarkable progress in capturing spatio-temporal dynamics, greatly advancing applications such as video understanding and editing.

The core technical foundation behind video Transformers is the spatio-temporal attention. In this setting, attention not only works on spatial patches within individual frames but also across temporal frames, thereby capturing spatio-temporal attention. Specifically, if the video contains $f$ frames and each frame is represented by $n$ tokens, then the full sequence has length $N = nf$. Computing the attention matrix requires $O(N^2)$ operations, and it is precisely this quadratic complexity that constitutes the fundamental bottleneck in video attention. As videos typically comprise hundreds of frames, $N$ is often larger than in text or image tasks, rendering the attention cost prohibitively high.

To address this challenge, previous work has investigated effective and approximate attention mechanisms in video modeling. Axial and variants restrict attention to the spatial or temporal dimensions (Wang et al., 2020), while other approaches approximate the attention matrix, such as Maxvit (Tu et al., 2022). These efforts have achieved subquadratic complexity in specific conditions, but they largely focus on static attention, in which the attention matrix is computed once for a fixed sequence.

However, video generation involves highly dynamic spatio-temporal structures, with attention weights evolving as new frames are generated. Recomputing the full attention matrix at every step is prohibitively expensive. Therefore, we propose a dynamic algorithm that is different from static approximation (Zandieh et al., 2023; Alman & Song, 2023), specifically for block-structured video attention.

**Definition 1.1** (Attention). *Suppose we have $Q, K, V \in \mathbb{R}^{n \times d}$, the static attention is defined by*

$$\mathsf{Attn}(Q, K, V) := D^{-1}AV,$$

*where $A \in \mathbb{R}^{n \times n}$ and diagonal matrix $D \in \mathbb{R}^{n \times n}$ is defined as*

$$A := \exp(QK^\top), \quad D := \mathrm{diag}(A\mathbf{1}_n).$$

But videos are not 1D. Each token now has a temporal index $[f]$ (frame) and a spatial index $n$ (patch inside the frame). Flattening $[f] \times [n]$ into a single axis will destroy the product structure and treats spatial and temporal neighborhoods as equally distant once they are far in the flattened order. Block attention, on the other hand, respects the natural product index set $[f] \times [n]$ by grouping tokens by frame.

**Definition 1.2** (Block attention). *Let the number of blocks be $f \in \mathbb{Z}_+$, each block with size $n \times d$, and total length $N := fn$.*

*Suppose we have $f$ query matrices $Q_1, Q_2, \cdots, Q_f \in \mathbb{R}^{n \times d}$, $f$ key matrices $K_1, K_2, \cdots, K_f \in \mathbb{R}^{n \times d}$, and $f$ value matrices $V_1, V_2, \cdots, V_f \in \mathbb{R}^{n \times d}$. Let $V := [V_1 \cdots V_f] \in \mathbb{R}^{N \times d}$.*

*Let $(i,j) \in [f] \times [f]$. We use $\widehat{A}_{i,j}$ to denote the element on $i$-th row and $j$-th column, and $\widehat{A}_{[i,j]}$ to denote the block on $i$-th row and $j$-th column, i.e., $\widehat{A}_{[i,j]} := \widehat{A}_{(i-1)n+[n],(j-1)n+[n]} \in \mathbb{R}^{n \times n}$. Then, the block attention computation is defined by*

$$\mathsf{BAttn}(\{Q_i, K_i, V_i\}_{i=1}^{f}) := \widehat{D}^{-1} \widehat{A} \widehat{V},$$

*where diagonal matrix $\widehat{D} \in \mathbb{R}^{N \times N}$ and matrix $\widehat{A} \in \mathbb{R}^{N \times N}$ are defined by*

$$\widehat{D} := \mathrm{diag}(\widehat{A} \mathbf{1}_N), \quad \widehat{A}_{[i,j]} := \exp(Q_i K_j^\top) \in \mathbb{R}^{n \times n}.$$

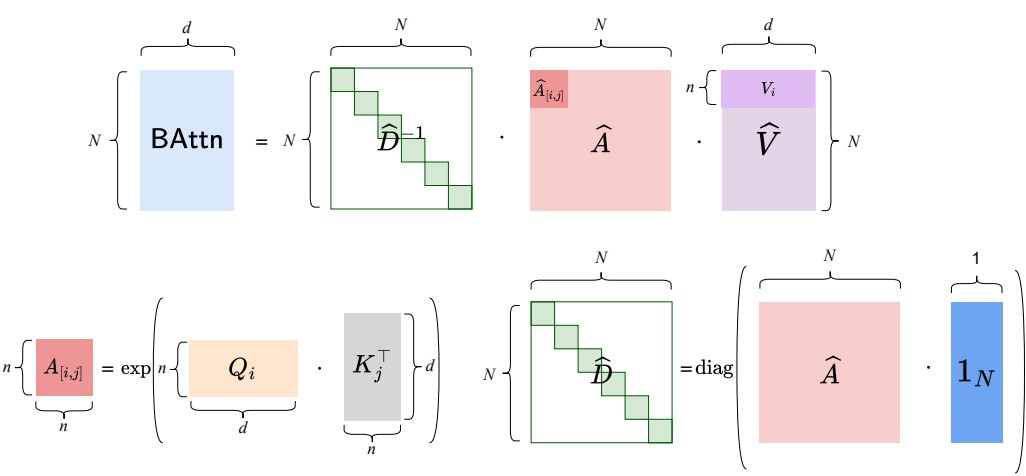

Figure 1: Computation of the $\mathsf{BAttn}(\{Q_i, K_i, V_i\}_{i=1}^{f})$ (Definition 1.2).

Intuitively, each frame $i$ distributes its attention mass over all frames $j$ and their spatial patches; within-frame (spatial) and cross-frame (temporal) interactions are captured by different blocks of $H$.

Naively computing the block attention yields $O(N^2) = O(n^2 f^2)$ time. Prior structured variants (e.g., axial/radial patterns) achieve subquadratic time only when one axis is much smaller than the other, leaving a wide parameter regime where full $O(n^2 f^2)$ persists.

An interesting property of videos is that videos exhibit strong temporal coherence: between adjacent frames, most spatial tokens persist with small changes, while global photometric effects (e.g., exposure/pan) act nearly low-dimensional. We utilize these properties, and formalize those by the following two assumptions:

**Assumption 1.3** (Difference between $K_i$). *For all the $\{K_i\}_{i \in [f-1]}$, we assume that $K_i$ and $K_{i+1}$ only different by a single row.*

**Assumption 1.4** (Rank-1 $k$-sparse changes in $V_i$). *For all the $\{V_i\}_{i \in [f-1]}$, we assume $V_i$ and $V_{i+1}$ has rank-1 $k$-sparse changes.*

Inspired by (Brand et al., 2024), we propose a dynamic algorithm that could compute fast block attention under above assumptions.

**Theorem 1.5** (Fast block attention). *For any constant $a \in (0,1]$. Let $d = O(n)$. Under Assumption 1.3 and Assumption 1.4, where $k = O(1)$. Then, there is a dynamic data structure that uses $O(fn^2)$ space and supports the following operations:*

- INIT($\{Q_i \in \mathbb{R}^{n \times d}, K_i \in \mathbb{R}^{n \times d}, V_i \in \mathbb{R}^{n \times d}\}_{i=1}^{f}$). *It runs in $O(f \cdot \mathcal{T}_{\mathrm{mat}}(n, d, n))$ time.*

- QUERY($x \in [n], y \in [n]$). *This operation outputs $\mathsf{BAttn}(\{Q_i, K_i, V_i\}_{i=1}^{f})_{x,y}$, which is defined in Definition 1.2, and takes $O(\mathcal{T}_{\mathrm{mat}}(n, n, n^a)\frac{f}{n^a})$ amortized time.*

**Roadmap.** In Section 2, we present the preliminary for our work. In Section 3, we review related work. In Section 4, we introduce fast block attention algorithms in our work. Finally, we provide the conclusion of our work in Section 5.

## 2 PRELIMINARY

**Notation**. We use $\mathbb{R}$ to denote the set of real number. We use $A_{i,j}$ to denote the element on $i$-th row and $j$-th column, and $A_{[i,j]}$ to denote the block on $i$-th row and $j$-th column, i.e., $A_{[i,j]} := A_{(i-1)n+[n],(j-1)n+[n]} \in \mathbb{R}^{n \times n}$. For any positive integer, we use $[n]$ to denote set $\{1, 2, \cdots, n\}$. We use $\mathbf{1}_n$ to denote a length $n$ vector whose entries are all ones. For any matrix $A \in \mathbb{R}^{m \times n}$, we use $\exp(A) \in \mathbb{R}^{m \times n}$ to denote entry-wise exponential function, i.e. $\exp(A)_{i,j} = \exp(A_{i,j})$. We use $\mathcal{T}_{\mathrm{mat}}(a, b, c)$ to denote the time to multiply an $a \times b$ matrix with a $b \times c$ matrix. Practical bound $\mathcal{T}_{\mathrm{mat}}(a, b, c) \leq O(abc)$. We use $\circ$ to denote entry-wise product.

### 2.1 BLOCK ATTENTION

We first provide a handful tool for block attention calculation.

**Lemma 2.1** (Block attention normalization). *Let $\widehat{D}$ defined by Definition 1.2. Then, $\widehat{D}_{[i,i]} = \mathrm{diag}(A_{i,*}\mathbf{1}_N)$.*

*Proof.*

$$\widehat{D} = \mathrm{diag}(A\mathbf{1}_N)$$
$$\widehat{D}_{[i,i]} = \mathrm{diag}(A\mathbf{1}_N)_{[i,i]}$$
$$= \mathrm{diag}(A_{i,*}\mathbf{1}_N),$$

where the first step follows from Definition 1.2, the second step take the $[i,i]$ sub-matrix, and the third step from basic algebra. $\square$

Then, we provide a simple algebra fact which is the core of our dynamic algorithm.

**Fact 2.2** (Folklore). *Given a set of vectors $a_1, \cdots, a_k \in \mathbb{R}^n$, and $b_1, \cdots, b_k \in \mathbb{R}^n$. If $A = [a_1 \cdots a_k], B = [b_1 \cdots b_k]$, then we have*

$$\sum_{i=1}^{k} a_i b_i^{\top} = AB.$$

### 2.2 DYNAMIC ATTENTION

Here, we present the main theorem of (Brand et al., 2024). The dynamic algorithm of (Brand et al., 2024) support fast computation for regular attention.

**Lemma 2.3** (Dynamic attention, Theorem 4.1 of (Brand et al., 2024)). *For any constant $a \in (0,1]$. Let $d = O(n)$. Then, there is a dynamic data structure that uses $O(n^2)$ space and supports the following operations:*

- INIT($Q, K, V$). *It runs in $O(\mathcal{T}_{\mathrm{mat}}(n, d, n))$ time.*

- UPDATEK$(i \in [n], j \in [d], \delta \in \mathbb{R})$. *This operation updates one entry in $K$, and it runs in $O(\mathcal{T}_{\mathrm{mat}}(n, n^a, n)/n^a)$ amortized time.*

- UPDATEV$(i \in [n], j \in [d], \delta \in \mathbb{R})$. *This operation takes same amortized time as $K$ update.*

- QUERY$(i \in [n], j \in [d])$. *This operation outputs* $\mathsf{Attn}(Q, K, V)_{i,j}$, *which is defined in Definition 1.1, and takes $O(n^a)$ worst-case time.*

## 3 RELATED WORK

**Attention.** Since the introduction of attention mechanisms in natural language processing (Vaswani et al., 2017), it has become the core architecture in a wide range of fields, including text (Devlin et al., 2019b; Brown et al., 2020), vision (Liu et al., 2021), and multimodality (Kim et al., 2021). In computer vision, early work such as ViT (Dosovitskiy et al., 2020) demonstrated that self-attention models could effectively handle image block sequences. With continuous development, including TimeSformer (Bertasius et al., 2021), extend the Transformer to video understanding tasks and propose a time-space decomposition attention mechanism. Additionally, ViViT (Arnab et al., 2021) extends Vision Transformer to video by factorizing spatial and temporal dimensions, while Video Swin Transformer (Liu et al., 2022) leverages shifted windows to capture spatio-temporal dependencies efficiently.

In a different recent work (Alman & Song, 2023), they focus on the bounded-entry setting with $d = O(\log n)$ and establish conditional lower bounds for approximating attention under the assumption that $\|Q\|_\infty, \|K\|_\infty, \|V\|_\infty \leq B$. Building on this line, (Alman & Song, 2024) generalizes softmax attention to Kronecker computation. Additionally, (Brand et al., 2024) investigates dynamic attention maintenance, giving conditional hardness results and optimal update algorithms with amortized complexity $O(n^{\omega(1,1,\tau)-\tau})$ under sparsity assumptions. More recently, (Alman & Song, 2025a) proposes Fast RoPE attention, which embeds the rotary positional into polynomial evaluation, and obtains running time $O(\log n)$ instead of $O(n^2)$. In another recent work, (Alman & Song, 2025b) proves that only sufficiently large weights, rather than skip connections, can prevent rank collapse. Despite the notable success of prior work, the quadratic computational cost of basic self-attention mechanisms remains a fundamental constraint, particularly in video generation tasks where the number of tokens far exceeds that in image or text scenarios. This challenge has motivated research on efficient attention mechanisms. To address this limitation, we investigate a structured video attention mechanism that leverages the block structure inherent in the spatio-temporal token to accelerate attention computation.

**Dynamic Algorithm.** Recently, dynamic algorithms have been widely studied in theoretical computer science, with the core objective being to design data structures that efficiently preserve certain properties of dynamically changing inputs. An outstanding example is projection maintenance, which has been applied in convex optimization to preserve the projection matrix (Jiang et al., 2020; van den Brand, 2021; Dong et al., 2021; Huang et al., 2022). Under these conditions, the projection matrix $P = B^\top (BB^\top)^{-1} B$ is typically maintained under low-rank or sparse updates of matrix $B \in \mathbb{R}^{m \times n}$. In contrast, our problem of computing attention matrices in video modeling exhibits two key differences. The first key distinction lies in the type of matrix inversion involved. In attention mechanisms, we focus on computing the inverse of a positive diagonal matrix, while in optimization tasks, the focus is often on the inverse of a full-rank matrix. The second major difference is that the attention mechanism innovates element nonlinearity, such as softmax, making it impractical to directly reuse linear update techniques. Concretely, when $f$ is linear, computing $f(QK^\top)V$ can be simplified by precomputing $K^\top V$. However, when $f$ is the exponential function, this simplification is no longer valid, and $K^\top V$ cannot be directly computed.

Motivated by these limitations, we propose a dynamic block attention algorithm. By dividing the spatio-temporal tokens into blocks, our approach maintains precise attention computations while leveraging an efficient update algorithm. This enables scalable attention processing for lengthy videos, overcoming computational bottlenecks and challenges posed by evolving attention matrices.

**Video Generation.** In recent years, video generation has developed rapidly. Early approaches relied on recurrent networks or 3D convolutions to capture spatio-temporal dynamics (Vondrick et al., 2016; Kalchbrenner et al., 2017), but these methods often generate temporally inconsistent

results and present limited scalability. With the advent of GAN-based approaches, methods such as MoCoGAN (Tulyakov et al., 2018) and TGAN (Saito et al., 2017) divided the visual signals of video into content and motion to improve video generation, though training stability and long-term coherence remained challenging. More recently, diffusion models have achieved state-of-the-art performance in video generation. For instance, Video Diffusion Models (Ho et al., 2022b) extend image diffusion to the temporal dimension, while architectures like Imagen Video (Ho et al., 2022a) and Make-A-Video (Singer et al., 2023) leverage large-scale text and video datasets for text-to-video generation.

A central component of these models is the utilisation of attention mechanisms, which enable long-term temporal modeling. For example, TimeSformer (Bertasius et al., 2021) applied self-attention for video understanding, while CogVideo (Hong et al., 2022) and Phenaki (Villegas et al., 2023) leverage attention for high-fidelity text-to-video generation. Despite these advances, computing attention remains computationally expensive in the video domain due to $O(N^2)$ quadratic cost across spatial and temporal dimensions. Therefore, our work proposes a block-based video attention algorithm to address the high computational cost.

## 4 FAST BLOCK ATTENTION

In this section, we introduce a dynamic data structure for fast block attention computation.

*Proof.* It trivially follows from Lemma 4.2, Lemma 4.3, Lemma 4.1. $\qquad\square$

---

**Algorithm 1** Dynamic Data Structure for Fast Block Attention

---

1: **data structure** FASTBLOCKATTENTION $\qquad\qquad\qquad\qquad\qquad\qquad\qquad\triangleright$ Theorem 1.5
2: **members**
3: $\quad Q_{[f]} \in \mathbb{R}^{n \times d}$ $\qquad\qquad\qquad\qquad\qquad\qquad\qquad\qquad\qquad\qquad\triangleright$ Query token
4: $\quad K_{[f]} \in \mathbb{R}^{n \times d}$ $\qquad\qquad\qquad\qquad\qquad\qquad\qquad\qquad\qquad\qquad\triangleright$ Key token
5: $\quad V_{[f]} \in \mathbb{R}^{n \times d}$ $\qquad\qquad\qquad\qquad\qquad\qquad\qquad\qquad\qquad\qquad\triangleright$ Value token
6: $\quad M_{[f]} \in \mathbb{R}^{n \times n}$ $\qquad\qquad\qquad\qquad\qquad\qquad\qquad\triangleright$ The logits matrix, $M = QK^\top$
7: $\quad A_{[f]} \in \mathbb{R}^{n \times n}$ $\qquad\qquad\qquad\qquad\qquad\triangleright$ The attention matrix, $A = \exp(QK^\top)$
8: $\quad D_{[f]} \in \mathbb{R}^{n \times n}$ $\qquad\qquad\qquad\qquad\qquad\qquad\qquad\triangleright$ The diagonal matrix,
9: $\quad C_{[f]} \in \mathbb{R}^{n \times d}$ $\qquad\qquad\qquad\qquad\triangleright$ Intermediate matrix, $C = \exp(QK^\top)V$
10: $\quad B_{[f]} \in \mathbb{R}^{n \times d}$ $\qquad\qquad\triangleright$ Attention matrix for a block, $B = D^{-1}AV$
11: $\quad \text{List}_{A,[f]}$ $\qquad\qquad\qquad\qquad\qquad\qquad\qquad\qquad\qquad\triangleright$ List with size $n^a$
12: $\quad \text{List}_{C,[f]}$ $\qquad\qquad\qquad\qquad\qquad\qquad\qquad\qquad\qquad\triangleright$ List with size $n^a$
13: $\quad \text{List}_{D,[f]}$ $\qquad\qquad\qquad\qquad\qquad\qquad\qquad\qquad\qquad\triangleright$ List with size $n^a$
14: $\quad \text{ct}_{K,[f]}, \text{ct}_{V,[f]}$ $\qquad\qquad\qquad\qquad\qquad\triangleright$ Counter for updates of $K$ and $V$
15: **end members**
16:
17: **procedure** INIT($\{Q_i, K_i, V_i\}_{i=1}^f$) $\qquad\qquad\qquad\qquad\qquad\qquad\triangleright$ Lemma 4.1
18: $\quad$ **for** $i \in [f]$ **do**
19: $\qquad Q_i \leftarrow Q_i, K_i \leftarrow K_i, V_i \leftarrow V_i$
20: $\qquad M_i \leftarrow Q_i K_1^\top, A_i \leftarrow \exp(Q_i K_1^\top)$
21: $\qquad C_i \leftarrow \exp(Q_i K_1^\top)V_1$
22: $\qquad A \leftarrow \exp(Q_i K_1^\top)$
23: $\qquad D \leftarrow \text{diag}(A\mathbf{1}_n).$
24: $\qquad B_i \leftarrow D^{-1}AV_1$
25: $\qquad \text{ct}_{K,i} \leftarrow 0$
26: $\qquad \text{ct}_{V,i} \leftarrow 0$
27: $\quad$ **end for**
28: **end procedure**
29: **end data structure**

---

---

**Algorithm 2** Query the block attention

---

1: **procedure** QUERY$(x, y)$                     ▷ Lemma 4.2, Lemma 4.3
2:     $i \leftarrow \lceil \frac{x}{n} \rceil$                                   ▷ Identify the row
3:     **for** $j \in \{2 \cdots \lfloor \frac{y}{n} \rfloor\}$ **do**
4:         $r \leftarrow$ the different row between $K_j$ and $K_{j-1}$         ▷ Assumption 1.3
5:         $\delta_K \leftarrow e_r^\top (K_j - K_{j-1})$
6:         UPDATEK$(i, r, \delta_K)$
7:         $\Delta_{V,1}, \Delta_{V,2} \leftarrow$ rank-1 difference between $V_j$ and $V_{j-1}$     ▷ Assumption 1.4
8:         UPDATEV$(i, \Delta_{V,1}, \Delta_{V,2})$
9:     **end for**
10:    **for** $j \in \{\lfloor \frac{y}{n} \rfloor \cdots f\}$ **do**
11:        $r \leftarrow$ the different row between $K_j$ and $K_{j-1}$         ▷ Assumption 1.3
12:        $\delta_K \leftarrow e_r^\top (K_j - K_{j-1})$
13:        UPDATED$(i, r, \delta_K)$
14:    **end for**
15:    Let $\Delta_{V,1}$ and $\Delta_{V,2}$ be rectangular matrix obtained from list from $\mathrm{List}_V$
16:    Let $(D_{\mathrm{tmp}})^{-1}$ denote the list of diagonal matrices obtained from $\mathrm{List}_D[\mathrm{ct}_K].\mathrm{GETB}$
17:    $\mathrm{answer}_1 \leftarrow (D_{\mathrm{tmp}})_x^{-1}(C + (\Delta_{C,1}\Delta_{C,2})_{x,y})$
18:    $\mathrm{answer}_2 \leftarrow (D_{\mathrm{tmp}})_x^{-1} A_{x,*} \Delta_{V,1}(\Delta_{V,2})_{*,x}$
19:    $\mathrm{answer} \leftarrow \sum_{j=1}^2 \mathrm{answer}_j$
20:    **return** answer
21: **end procedure**
22: **end data structure**

---

**Algorithm 3** Algorithm that update $K$ and maintain the data structure

---

1: **data structure** FASTBLOCKATTENTION                     ▷ Theorem 1.5
2: **procedure** UPDATEK$(i \in [n], \delta_K \in \mathbb{R}^d)$               ▷ Lemma 4.4
3:    */\*For all members in the data structure, we omit the $i$ subscript for simplicity\*/*
4:    $\mathrm{ct}_K \leftarrow \mathrm{ct}_K + 1$
5:    $\widetilde{K}_{i,*} \leftarrow K_{i,*} + \delta_K^\top$
6:    $(\Delta_M)_{*,i} \leftarrow \underbrace{Q}_{n \times d} \underbrace{\delta_K}_{d \times 1}$             ▷ $\Delta_M$ only have entries in $i$-th column
7:    $(\Delta_A)_{*,i} \leftarrow (A_{*,i} \circ (\exp((\Delta_M)_{*,i}) - \mathbf{1}_n))$
8:    $\widetilde{M} \leftarrow M + (\Delta_M)_{*,i} e_i^\top$            ▷ We only update $i$-th column of $M$
9:    $\widetilde{A} \leftarrow A + (\Delta_A)_{*,i} e_i^\top$             ▷ We only update $i$-th column of $A$
10:   Obtain diagonal vector $D_{\mathrm{tmp}}$ from $\mathrm{List}_D[\mathrm{ct}_K - 1].\mathrm{GETB}$     ▷ It takes $O(n)$ time
11:   $\widetilde{D} \leftarrow D_{\mathrm{tmp}}^{-1} + \mathrm{diag}(\Delta_A)_{*,i}$
12:   **for** $j = 1 \to n$ **do**
13:       $(\Delta_D)_{j,j} \leftarrow (D_{\mathrm{tmp}})_{j,j}^{-1} - \widetilde{D}_{j,j}^{-1}$
14:   **end for**
15:   **if** $\mathrm{ct}_K < n^a$ **then**
16:       $\mathrm{List}_C[\mathrm{ct}_K - 1].(a, b) \leftarrow ((\Delta_A)_{*,i} \in \mathbb{R}^n, V^\top e_i \in \mathbb{R}^d)$
17:       $\mathrm{List}_D[\mathrm{ct}_K - 1].(a, b) \leftarrow (\Delta_D \in \mathbb{R}^{n \times n}, \widetilde{D}^{-1} \in \mathbb{R}^{n \times n})$    ▷ Diagonal matrices
18:   **else**                                    ▷ $\mathcal{T}_{\mathrm{mat}}(n, n^a, d) = n^{\omega(1,1,a)}$ time
19:       RECOMPUTE()                  ▷ Algorithm 6. Re-compute everything
20:   **end if**
21:   */\*Referesh the memory\*/*
22:   $K \leftarrow \widetilde{K}$
23:   $A \leftarrow \widetilde{A}$
24:   $M \leftarrow \widetilde{M}$
25: **end procedure**
26: **end data structure**

---

---

**Algorithm 4** Algorithm that update $D$ and maintain the data structure

1: **data structure** FASTBLOCKATTENTION           ▷ Theorem 1.5
2:   **procedure** UPDATED($i \in [n], \delta_K \in \mathbb{R}^d$)        ▷ Lemma 4.5
3:    /*For all members in the data structure, we omit the $i$ subscript for simplicity*/
4:    $\text{ct}_K \leftarrow \text{ct}_K + 1$
5:    $\widetilde{K}_{i,*} \leftarrow K_{i,*} + \delta_K^\top$
6:    $(\Delta_M)_{*,i} \leftarrow \underbrace{Q}_{n \times d} \underbrace{\delta_K}_{d \times 1}$        ▷ $\Delta_M$ only have entries in $i$-th column
7:    $(\Delta_A)_{*,i} \leftarrow (A_{*,i} \circ (\exp((\Delta_M)_{*,i}) - \mathbf{1}_n))$
8:    $\widetilde{M} \leftarrow M + (\Delta_M)_{*,i} e_i^\top$        ▷ We only update $i$-th column of $M$
9:    Obtain diagonal vector $D_{\text{tmp}}$ from $\text{List}_D[\text{ct}_K - 1].\text{GETB}$    ▷ It takes $O(n)$ time
10:    $\widetilde{D} \leftarrow D_{\text{tmp}}^{-1} + \text{diag}(\Delta_A)_{*,i}$
11:    **for** $j = 1 \to n$ **do**
12:     $(\Delta_D)_{j,j} \leftarrow (D_{\text{tmp}})_{j,j}^{-1} - \widetilde{D}_{j,j}^{-1}$
13:    **end for**
14:    **if** $\text{ct}_K < n^a$ **then**
15:     $\text{List}_D[\text{ct}_K - 1].(a, b) \leftarrow (\Delta_D \in \mathbb{R}^{n \times n}, \widetilde{D}^{-1} \in \mathbb{R}^{n \times n})$   ▷ Diagonal matrices
16:    **else**            ▷ $\mathcal{T}_{\text{mat}}(n, n^a, d) = n^{\omega(1,1,a)}$ time
17:     RECOMPUTE()        ▷ Algorithm 6. Re-compute everything
18:    **end if**
19:    /*Referesh the memory*/
20:    $K \leftarrow \widetilde{K}$
21:    $M \leftarrow \widetilde{M}$
22: **end procedure**
23: **end data structure**

---

**Algorithm 5** Algorithm that update $V$ and maintain the data structure

1: **data structure** FASTBLOCKATTENTION           ▷ Theorem 1.5
2:   **procedure** UPDATEV($\delta_{V,1} \in \mathbb{R}^n, \delta_{V,2} \in \mathbb{R}^d$)      ▷ Lemma 4.6
3:    /*For all members in the data structure, we omit the $i$ subscript for simplicity*/
4:    $\text{ct}_V \leftarrow \text{ct}_V + 1$
5:    **if** $\text{ct}_V < n^a$ **then**
6:     $\text{List}_V[\text{ct}_V - 1].(a, b) \leftarrow (\delta_{V,1}, \delta_{V,2})$
7:    **else**
8:     RECOMPUTE()        ▷ Algorithm 6. Re-compute everything
9:    **end if**
10: **end procedure**
11: **end data structure**

---

### 4.1 INIT

We begin by stating the running time of the initialization procedure.

**Lemma 4.1** (Running time of INIT). *The running time of procedure* INIT *(Algorithm 1) is* $O(f \cdot \mathcal{T}_{\text{mat}}(n, d, n))$.

*Proof.* It is trivially from applying fast matrix multiplication $f$ iterations.     □

### 4.2 QUERY

Next, we show the running time of QUERY.

**Lemma 4.2** (Running time of QUERY). *The running time of procedure* QUERY *(Algorithm 2) is* $O(\mathcal{T}_{\text{mat}}(n, n, n^a) \frac{f}{n^a})$.

*Proof.* **Part 1.** UPDATEK and UPDATEV. The amortized running time is $O(\mathcal{T}_{\text{mat}}(n, n, n^a) \frac{f}{n^a})$.

---

**Algorithm 6** Algorithm that re-compute evreything

---

1: **data structure** FASTBLOCKATTENTION                                    ▷ Theorem 1.5
2: **procedure** RECOMPUTE($i \in [n]$)                              ▷ Lemma A.1, Lemma A.2
3:    /*For all members in the data structure, we omit the $i$ subscript for simplicity*/
4:    Let $\Delta_{C,1}$ and $\Delta_{C,2}$ be rectangular matrix obtained from $\mathrm{List}_C$
5:    Let $\Delta_{V,1}$ and $\Delta_{V,2}$ be rectangular matrix obtained from $\mathrm{List}_V$
6:    Let $\Delta_D(i)$ denote the list of diagonal matrices obtained from $\mathrm{List}_D[i].\mathrm{GETA}$
7:    $\widetilde{C} \leftarrow C + \Delta_{C,1} \cdot \Delta_{C,2}$                    ▷ It takes $\mathcal{T}_{\mathrm{mat}}(n, n^a, d)$ time
8:    $\widetilde{V} \leftarrow V + \Delta_{V,1} \cdot \Delta_{V,2}$                    ▷ It takes $\mathcal{T}_{\mathrm{mat}}(n, n^a, d)$ time
9:    $\Delta_D \leftarrow \sum_{i=1}^{\mathrm{ct}_K} \Delta_D(i)$                           ▷ it takes $n^{1+a}$ time
10:   $\widetilde{D}^{-1} \leftarrow D^{-1} + \Delta_D$                                  ▷ It takes $n$ time
11:   $\widetilde{B} \leftarrow \widetilde{D}^{-1} \cdot \widetilde{C}$                                ▷ This takes $nd$
12:   /*Refresh the memory*/
13:   $D \leftarrow \widetilde{D}, C \leftarrow \widetilde{C}, B \leftarrow \widetilde{B}, V \leftarrow \widetilde{V}$
14:   /*Reset the counter*/
15:   $\mathrm{ct}_K \leftarrow 0, \mathrm{ct}_V \leftarrow 0$
16: **end procedure**
17: **end data structure**

---

**Part 2.** UPDATED. The amortized running time is $O(\mathcal{T}_{\mathrm{mat}}(n, n, n^a)\frac{f}{n^a})$.

**Part 3.** We first stack all the vectors in $\mathrm{List}_V$ to $\Delta_{V,1}$ and $\Delta_{V,2}$, which takes $O(1)$ time.

Computing $(D_{\mathrm{tmp}})_i^{-1}(C + \Delta_{C,1}\Delta_{C,2})_{i,j}$ takes $O(n^a)$ time.

Computing $\Delta_{V,1}\Delta_{V,2}$ takes $O(n^a)$ time as $\Delta_{V,1}$ and $\Delta_{V,2}$ are $k$-sparse (Assumption 1.4).

Computing $(D_{\mathrm{tmp}})_i^{-1}A_{i,*}(\Delta_{V,1}\Delta_{V,2})_{*,j}$ takes $O(n^a)$ time as $\mathrm{nnz}((\Delta_{V,1}\Delta_{V,2})_{*,j}) = O(1)$.

Hence the amortized running time is $O(\mathcal{T}_{\mathrm{mat}}(n, n, n^a)\frac{f}{n^a})$                    □

Now, we establish the correctness of QUERY.

**Lemma 4.3** (Correctness of QUERY). *The procedure* QUERY$(x, y)$ *(Algorithm 2) outputs*

$$\mathrm{BAttn}(\{Q_i, K_i, V_i\}_{i=1}^f)_{x,y}.$$

*Proof.* Let $(D_{\mathrm{tmp}})^{-1}$ denote the diagonal matrix obtained from $\mathrm{List}_D[\mathrm{ct}_K].\mathrm{GETB}$. Let $\Delta_{V,1}$ and $\Delta_{V,2}$ be rectangular matrix obtained from $\mathrm{List}_V$.

Since we know

$$\mathrm{answer}_1 = (D_{\mathrm{tmp}})_x^{-1}(C + \Delta_{C,1}\Delta_{C,2})_{x,y},$$

and

$$\mathrm{answer}_2 = (D_{\mathrm{tmp}})_x^{-1}A_{x,*}(\Delta_{V,1}\Delta_{V,2})_{*,x}.$$

By summing them up, we get

$$\begin{aligned}
\widetilde{B} &= \mathrm{answer}_1 + \mathrm{answer}_2 \\
&= (D_{\mathrm{tmp}})_x^{-1}(C + \Delta_{C,1}\Delta_{C,2})_{x,y} + (D_{\mathrm{tmp}})_x^{-1}A_{x,*}(\Delta_{V,1}\Delta_{V,2})_{*,y} \\
&= (D_{\mathrm{tmp}})_x^{-1}(AV)_{x,y} + (D_{\mathrm{tmp}})_x^{-1}A_{x,*}(\Delta_{V,1}\Delta_{V,2})_{*,y} \\
&= (D_{\mathrm{tmp}})_x^{-1}(AV)_{x,y} + (D_{\mathrm{tmp}})_x^{-1}(A\Delta_{V,1}\Delta_{V,2})_{x,y} \\
&= (D_{\mathrm{tmp}})_x^{-1}(A(V + \Delta_{V,1}\Delta_{V,2}))_{x,y} \\
&= (D_{\mathrm{tmp}})_x^{-1}(AV_j)_{x,y} \\
&= \mathrm{diag}(\exp(Q_iK_j^\top)_{i,*}\mathbf{1}_N)_x^{-1}(\exp(Q_iK_j^\top)V_j)_{x,y} \\
&= (\mathrm{diag}(\exp(Q_iK_j^\top)_{i,*}\mathbf{1}_N)^{-1}\exp(Q_iK_j^\top)V_j)_{x,y}
\end{aligned}$$

$$= \mathsf{BAttn}(\{Q_i, K_i, V_i\}_{i=1}^f)_{x,y},$$

where the first step combines two answers, the second step substitute for the two answers, the third step follows from the definition of $C$ and UPDATEK (Algorithm 3), the fourth and the fifth steps follow from basic algebra, the sixth step follows from definition of $\Delta_{V,1}$, $\Delta_{V,2}$ and Fact 2.2, the seventh step follows from UPDATED (Algorithm 4), the eighth step follows from basic algebra, and the last step follows from Lemma 2.1 and Definition 1.2. □

### 4.3 UPDATE $K$, $V$ AND $D$

First, we show the running time of updating $K$.

**Lemma 4.4** (Running time of UPDATEK, informal version of B.1). *The procedure* UPDATEK *(Algorithm 3) takes* **Part 1.** $\mathcal{T}_{\mathrm{mat}}(n, n, n^a)$ *time in the worst case.* **Part 2.** $\mathcal{T}_{\mathrm{mat}}(n, n, n^a)/n^a$ *time in the amortized case.*

Next, we give the running time of updating $D$.

**Lemma 4.5** (Running time of UPDATED). *The procedure* UPDATED *(Algorithm 4) takes* **Part 1.** $\mathcal{T}_{\mathrm{mat}}(n, n, n^a)$ *time in the worst case.* **Part 2.** $\mathcal{T}_{\mathrm{mat}}(n, n, n^a)/n^a$ *time in the amortized case.*

*Proof.* The proof trivially follows from Lemma B.1. □

Finally, we analyze the running time of updating $V$.

**Lemma 4.6** (Running time of UPDATEV). *The procedure* UPDATEV *(Algorithm 5) takes* **Part 1.** $\mathcal{T}_{\mathrm{mat}}(n, n, n^a)$ *time in the worst case.* **Part 2.** $\mathcal{T}_{\mathrm{mat}}(n, n, n^a)/n^a$ *time in the amortized case.*

*Proof.* The proof trivially follows from Lemma B.1. □

### 4.4 RECOMPUTE

We first give the running time of the recompute.

**Lemma 4.7** (Running time of RECOMPUTE, Lemma 4.9 of (Brand et al., 2024)). *The running time of procedure* RECOMPUTE *(Algorithm 6) is* $\mathcal{T}_{\mathrm{mat}}(n, n^a, d)$.

*Proof.* For given $i \in [n]$, the RECOMPUTE works same as RECOMPUTE in (Brand et al., 2024). Then, according to Lemma A.1, the procedure runs in $\mathcal{T}_{\mathrm{mat}}(n, n^a, d)$ time. □

Then, we present the correctness of the recompute.

**Lemma 4.8** (Correctness of RECOMPUTE, Lemma 4.8 of (Brand et al., 2024)). *The procedure* RECOMPUTE *(Algorithm 6) correctly re-compute* $D, C, B, V$.

*Proof.* For given $i \in [n]$, the RECOMPUTE works same as RECOMPUTE in (Brand et al., 2024). Then, according to Lemma A.2, the procedure is correct. □

## 5 CONCLUSION

In this work, we investigated the fundamental computational bottleneck of video attention and introduced a dynamic algorithm for fast block attention. By leveraging temporal coherence within videos for row-wise key updates and exploiting low-rank variations in value between consecutive frames, our algorithm achieves $O(\mathcal{T}_{\mathrm{mat}}(n, n, n^a)\frac{f}{n^a})$ running time, representing a significant reduction in computational cost compared to the original $O(n^2 f^2)$ approach. Our findings demonstrate that dynamic block attention not only enhances efficiency theoretically but also paves the way for new directions in scalable video modeling, particularly within generative settings where attention must be updated frame by frame.

## ETHIC STATEMENT

This paper does not involve human subjects, personally identifiable data, or sensitive applications. We do not foresee direct ethical risks. We follow the ICLR Code of Ethics and affirm that all aspects of this research comply with the principles of fairness, transparency, and integrity.

## REPRODUCIBILITY STATEMENT

We ensure reproducibility of our theoretical results by including all formal assumptions, definitions, and complete proofs in the appendix. The main text states each theorem clearly and refers to the detailed proofs. No external data or software is required.

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

# Appendix

**Roadmap.** In the appendix, we first list some result from (Brand et al., 2024) in Section A. Then, we provide some formal results of our algorithm in Section B.

## A   DYNAMIC ATTENTION

In this section, we present some lemmas of the dynamic algorithm in (Brand et al., 2024).

We first provide the running time of its RECOMPUTE.

**Lemma A.1** (Running time of RECOMPUTE, Lemma 4.9 of (Brand et al., 2024)). *The running time of procedure* RECOMPUTE *(Algorithm 6) is* $\mathcal{T}_{\mathrm{mat}}(n, n^a, d)$.

Then, we provide the correctness of RECOMPUTE.

**Lemma A.2** (Correctness of RECOMPUTE, Lemma 4.8 of (Brand et al., 2024)). *The procedure* RECOMPUTE *(Algorithm 6) correctly re-compute* $D, C, B, V$.

## B   FAST BLOCK ATTENTION

In this section, we present formal version of some lemmas for our dynamic algorithms.

First we show the formal lemma of running time of UPDATEK.

**Lemma B.1** (Running time of UPDATEK, formal version of 4.4). *The procedure* UPDATEK *(Algorithm 3) takes*

- $\mathcal{T}_{\mathrm{mat}}(n, n, n^a)$ *time in the worst case.*

- $\mathcal{T}_{\mathrm{mat}}(n, n, n^a)/n^a$ *time in the amortized case.*

*Proof.* **Part 1.** It trivially from Lemma A.1

**Part 2.** If the $\mathrm{ct}_K < n^a$, we pay $O(n)$ time. If $\mathrm{ct}_K = n^a$, we pay $n^{\omega(1,1,a)}$. So the amortized time is

$$\frac{n(n^a - 1) + n^{\omega(1,1,a)}}{n^a} = O(n^{\omega(1,1,a)-a})$$

Note that, by using fast matrix multiplication and the fact that $d = O(n)$, we have $n^{\omega(1,1,a)} = \mathcal{T}_{\mathrm{mat}}(n, n^a, d)$. Thus we complete the proof. $\square$

## LLM USAGE DISCLOSURE

LLMs were used only to polish language, such as grammar and wording. These models did not contribute to idea creation or writing, and the authors take full responsibility for this paper's content.

