# OpenReview forum: "Fast Block Attention Computation via Dynamic Algorithm"
_ICLR.cc/2026/Conference — ICLR 2026 Conference Withdrawn Submission_

### Official Review · Reviewer_vTcw · 2025-11-01

**Soundness:** 1
**Presentation:** 1
**Contribution:** 1
**Rating:** 2
**Confidence:** 4

**Summary:**

This paper tries to address the significant computational bottleneck of the spatio-temporal attention mechanism in video Transformer models. The standard attention mechanism becomes prohibitively expensive for long videos, where the total number of tokens (frames × patches per frame) is large. To tackle this challenge, the authors propose a novel dynamic algorithm specifically for block attention. The core idea is to leverage the strong temporal coherence typically found between adjacent frames in a video. This coherence is formalized into two key assumptions: (1) the key matrices of consecutive frames differ by only a single row, and (2) the changes in the value matrices are low-rank (specifically, rank-1) and sparse. Based on these assumptions, the paper introduces a dynamic data structure that can answer attention queries in O(T_mat(n, n, n^α)) amortized time, where α ∈ [0, 1). However, this paper does not include any experimental design and results.

**Strengths:**

1. The authors provide formalisms for block attention and explicitly state the assumptions their algorithm relies on.

**Weaknesses:**

1. **Complete Lack of Empirical Validation**: For a paper focused on improving computational efficiency, the total absence of experimental results is a major shortcoming. Even for a theoretical contribution, some form of empirical validation is expected to substantiate the claims.

This submission feels incomplete. Without any experimental support, it is difficult for the reviewer to reasonably assess the theoretical contributions or provide constructive feedback on them.

**Questions:**

Could the authors explain the decision to omit all experimental validation?

---

> ### Author Response · Authors · 2025-12-02
>
> Thank you for your thoughtful feedback. Your comments are very helpful and much appreciated. We will address these in the next version.

---

### Official Review · Reviewer_sAEE · 2025-11-02

**Soundness:** 2
**Presentation:** 3
**Contribution:** 2
**Rating:** 4
**Confidence:** 4

**Summary:**

The paper studies the computational bottleneck of video Transformers and proposes a dynamic data structure to compute block attention in subquadratic time, under the assumption that consecutive frames differ very slightly in keys (one-row change) and have low-rank, k-sparse changes in values. Building on the dynamic attention line of work, the authors extend the idea from flat sequences to a block-structured, spatio‑temporal setting, arguing that videos have strong temporal coherence so such updates are realistic. The main theorem shows that, with these assumptions and d = O(n), QUERY can be supported in O(T_mat(n, n, n^a) · f / n^a) amortized time while using O(fn²) space. The paper gives definitions, algorithms, and proofs, but does not provide empirical validation on real video models or compare with practical video-attention systems.

**Strengths:**

- The paper gives a clean formalization of block attention and shows how to maintain it dynamically, which is nontrivial because of softmax and because blocks interact across frames.
- The algorithm is fully specified and the space/time bounds are explicit, so the theoretical part is reproducible.
- The work connects video‑modeling practice with dynamic‑algorithm theory, which is a direction ICLR should see more of.
- Theoretical parts are internally consistent: the paper carefully restates block attention (Def. 1.2), gives the assumptions, and then layers the dynamic updates, reusing lemmas from prior dynamic attention work.

**Weaknesses:**

1. **Assumptions look engineered**: Assumption 1.3 (“two adjacent K differ by a single row”) and Assumption 1.4 (“V changes are rank‑1 k‑sparse”) are exactly what the algorithm needs; the paper provides no measurement on real video Q/K/V to show these are even approximately true. Without such evidence, the theorem risks being a solution to a synthetic problem.
2. **No empirical validation**: there is no experiment on real video Transformers, no ablation on how often updates happen, no runtime vs. sequence length curves, no comparison to axial/windowed/MaxViT‑style attention that is common in practice. This seriously limits the paper’s ICLR relevance.
3. **Scope of improvement is unclear**: the claimed complexity relies on d = O(n) and on small k; many practical models use d unrelated to spatial tokens, and video diffusion often breaks the “small change per frame” pattern.

**Questions:**

1. Do you have any empirical statistics from real video models to show how many rows of K actually change per frame in practice?
2. If more than one row changes in K, can your data structure degrade gracefully, or does it immediately lose the subquadratic property?
3. Can you show at least one wall‑clock experiment that your method beats a well‑tuned windowed attention on a realistic video length?
4. Is it possible to relax Assumption 1.4 to low‑rank (not k‑sparse) changes, which seem more realistic for global illumination/motion?

---

> ### Author Response · Authors · 2025-12-02
>
> Thank you for your thoughtful feedback. Your comments are very helpful and much appreciated. We will address these in the next version.

---

### Official Review · Reviewer_jBMX · 2025-11-03

**Soundness:** 1
**Presentation:** 2
**Contribution:** 2
**Rating:** 2
**Confidence:** 4

**Summary:**

This paper aims to speed up attention for video transformers. By utilization temporal coherence of videos, this paper introduces two assumptions about the Q/K matrices between adjacent frames. Based on those assumptions, this paper proposes a dynamic algorithm which reduce the time complexity of attention by a factor $f$ where $f$ is the number of frames.

This paper is a theoretical paper without experiments.

**Strengths:**

1. The mathematical notation is clear and consistent.
2. The authors build a specific data structure to speed up attention. I appreciate their efforts.

**Weaknesses:**

1. The proposed algorithm is based on two assumptions. However, the first assumption is non-sense to me. It says that **the K matrices between two adjacent frames only different by a single token**. Since this assumption is completely invalid, I have no interest in understanding the algorithmic details.

2. The representation should be improved.

a) Please use more ink to describe the core idea. Some algorithm tables can be moved to appendix.

b) The notation $T_{mat}$ is used without explanation.

c) In figure 1, the graph of $\hat{D}$ is confusing. It is a one diagonal matrix, instead of a block-wise  diagonal matrix.

3. It is necessary to verify the assumptions on real world dataset.

**Questions:**

-

---

> ### Author Response · Authors · 2025-12-02
>
> Thank you for your thoughtful feedback. Your comments are very helpful and much appreciated. We will address these in the next version.

---

### Official Review · Reviewer_y83s · 2025-11-06

**Soundness:** 1
**Presentation:** 1
**Contribution:** 1
**Rating:** 0
**Confidence:** 4

**Summary:**

This paper proposes a fast algorithm for computing block-structured video attention with reduced computational complexity. It provides only the complexity analysis without implementation results.

**Strengths:**

Developing fast algorithm for applying transformer to video is an important problem.

**Weaknesses:**

A) Novelty:
It extends an existing dynamic algorithm proposed for ordinary attention of Transformer to block-structured video attention. The contribution is incremental.

B) Paper Organization and Presentation:

The paper presentation and organization is problematic.

Definition 1.2 is hard to follow.
\hat{A}_{I,j} and \hat{A}_{[I,j]} are different. The notation \hat{A} is also used, and it is unclear which \hat{A} is being referred to. Also, n is different from [n]. The proposed notations are confusing. Clarify should be improved. Figure 1 helps.

H is used on the third line after Figure 1, but undefined.

Assumption 1.3: It is not clear why K_{i} and K_{i+1} are different by a single row. What is the implication?
Assumption 1.4: Justification and implication is also needed.

According to the current organization of the paper, it is a bit strange that Section 3 on related work appears in the middle of developing the problem to be tackled. It should be move to the beginning or to the end.

Section 4 starts with the pseudo code of five algorithms listed, followed by a number of lemma. The paper ends after that. I see this as an unfinished paper.

C) No experimental result are presented in the paper.

**Questions:**

The paper should be better articulated, explained and organized. Also, experimental results to support the complexity analysis is important.

---

> ### Author Response · Authors · 2025-12-02
>
> Thank you for your thoughtful feedback. Your comments are very helpful and much appreciated. We will address these in the next version.

---

### Note · Authors · 2025-12-02

**Comment:**

We would like to sincerely thank all the reviewers for providing insightful feedback. After careful consideration, we have decided to withdraw this paper.

**Withdrawal Confirmation:**

I have read and agree with the venue's withdrawal policy on behalf of myself and my co-authors.